# Left Atrium Assessment by Speckle Tracking Echocardiography in Cryptogenic Stroke: Seeking Silent Atrial Fibrillation

**DOI:** 10.3390/jcm10163501

**Published:** 2021-08-09

**Authors:** Mireia Ble, Begoña Benito, Elisa Cuadrado-Godia, Sílvia Pérez-Fernández, Miquel Gómez, Aleksandra Mas-Stachurska, Helena Tizón-Marcos, Lluis Molina, Julio Martí-Almor, Mercè Cladellas

**Affiliations:** 1Medicine Department, Universidad Autónoma de Barcelona, 08035 Barcelona, Spain; mcladellas@psmar.cat; 2Cardiology Department, Hospital del Mar, Parc de Salut Mar, 08003 Barcelona, Spain; amasstachurska@psmar.cat (A.M.-S.); HTizon@parcdesalutmar.cat (H.T.-M.); lmolina@psmar.cat (L.M.); JMarti@parcdesalutmar.cat (J.M.-A.); 3Hospital del Mar Medical Research Institute (IMIM), 08003 Barcelona, Spain; b.benito.v@gmail.com (B.B.); ECuadrado@parcdesalutmar.cat (E.C.-G.); silvia_medina2712@hotmail.com (S.P.-F.); 4Cardiology Department, Hospital Vall d’Hebron, 08035 Barcelona, Spain; 5Neurology Department, Hospital del Mar, Parc de Salut Mar, 08003 Barcelona, Spain; 6CIBER of Cardiovascular Diseases (CIBERCV), 08003 Barcelona, Spain; 7Cardiology Department, Hospital de Barcelona, 08034 Barcelona, Spain; miquelgomez@scias.com

**Keywords:** atrial fibrillation, echocardiography, atrial strain, atrial disease, cryptogenic stroke

## Abstract

Silent atrial fibrillation (AF) may be the cause of some cryptogenic strokes (CrS). The aim of the study was to analyse atrial size and function by speckle tracking echocardiography in CrS patients to detect atrial disease. Patients admitted to the hospital due to CrS were included prospectively. Echocardiogram analysis included left atrial ejection fraction (LAEF) and atrial strain. Insertable cardiac monitor was implanted, and AF was defined as an episode of ≥1 min in the first year after stroke. Left atrial enlargement was defined as indexed volume > 34 mL/m^2^. Seventy-five consecutive patients were included, aged 76 ± 9 years (arterial hypertension 75%). AF was diagnosed in 49% of cases. The AF group had higher atrial volume and worse atrial function: peak atrial longitudinal strain (PALs) 19.6 ± 5.7% vs. 29.5 ± 7.2%, peak atrial contraction strain (PACs) 8.9 ± 3.9% vs. 16.5 ± 6%, LAEF 46.8 ± 11.5% vs. 60.6 ± 5.2%; *p* < 0.001. AF was diagnosed in 20 of 53 patients with non-enlarged atrium, and in 18 of them, atrial dysfunction was present. The multivariate logistic regression analysis demonstrated an independent association between detection of AF and atrial volume, LAEF, and strain. Cut-off values were obtained: LAEF < 55%, PALs < 21.4%, and PACs < 12.9%. In conclusion, speckle tracking echocardiography in CrS patients improves silent atrial disease diagnosis, with or without atrial enlargement.

## 1. Introduction

Nowadays, stroke is one of the main causes of morbidity and mortality, leading to it being the second cause of death worldwide (World Health Organisation http://who.int/home-page/index.es.shtml (accessed on 8 July 2021)). Despite new technologies and the carrying out of accurate diagnostic procedures, between 20 and 30% of ischemic strokes are of unknown cause and currently are classified as being cryptogenic stroke (CrS) [1,2]. Several mechanisms can lead to CrS and, among these, cardioembolic stroke due to silent atrial fibrillation (AF) could be detected in up to 30% of cases [1,2].

Left atrium (LA) size is a strong predictor of cardioembolic stroke and AF [3,4]. However, the presence of atrial enlargement in other diseases such as diastolic dysfunction and arterial hypertension limits its interpretation to assess thromboembolic risk. Moreover, it has been described that the paroxysmal AF can also be present with normal LA size [5].

Recently, speckle tracking echocardiography has been widely accepted as a novel technology for evaluating mechanics, function, and LA remodelling useful in the prediction of AF [6,7]. Previous studies have shown a significant association between decreased LA strain and stroke [8,9], even in patients with low-risk CHADS_2_ score ≤ 1 [10].

AF in CrS can appear as a paroxysmal and asymptomatic form, which leads to underdiagnosed cases [11,12]. This has important implications due to the lack of appropriate secondary stroke prevention with anticoagulation. In this regard, continuous electrocardiographic monitoring has been shown to be superior to the intermittent or short recordings for detecting AF [13,14].

In spite of the proven usefulness of all these echocardiographic parameters and prolonged monitoring to detect AF, there are no studies that have evaluated both techniques together in patients with CrS. For this reason, the main objective of the study was to evaluate prospectively atrial anatomy and function using speckle tracking echocardiography in patients with CrS and detection of silent AF by an insertable cardiac monitor.

## 2. Materials and Methods

From October 2013 to September 2016, patients between 50 and 89 years old who were admitted to the Stroke Unit with CrS according to the SSS-TOAST criteria were prospectively included. Patients with permanent contraindication for oral anticoagulant treatment or indication for anticoagulation for other reasons, those whose life expectancy was lower than 1 year, and those with severe comorbidity or disabling stroke (modified Rankin scale > 4) were ruled out.

CrS diagnosis was in accordance with clinical practice guidelines [15] when magnetic resonance angiography or angioCT displayed a non-lacunar brain infarct that excluded extracranial and intracranial arterial stenosis or occlusion due to atherosclerosis, vasculitis, or dissection and ruling out a cardioembolic source. All patients underwent a comprehensive work-up during admission, including 12-lead ECG, transthoracic echocardiography, brain computed tomography, blood test, and neurovascular imaging (magnetic resonance angiography, angioCT, and/or 2D ultrasound of supra-aortic trunks and intracranial territory).

### 2.1. Patient Monitoring

All patients were continuously ECG-monitored for at least 48 h and underwent implant of an insertable cardiac monitor before hospital discharge. AF was defined as an episode ≥ 1 min in the first year after the stroke. The device used was the Biomonitor-2 (Biotronik^®^) that has specific algorithms for the detection of AF, defined by a variability of RR > 12.5%; bradycardia (<30 bpm), asystole (>3 s), and tachycardia events (>160 bpm) were also recorded, as well as episodes manually activated by the user in case of symptoms. The insertable cardiac monitor allows for reading and obtaining data daily by telematic message to the treating team (Biotronik Home Monitoring, HM). The insertable loop recorder was read and interpreted by cardiologists specialised in arrhythmias, without knowledge of echocardiographic results.

### 2.2. Echocardiographic Analysis of Atrial Anatomy and Function

Standard 2D transthoracic echocardiogram was performed with General Electric Vivid E9 equipment (GE Healthcare Vingmed, Trodheim, Norway) that included anatomical and functional assessment of the left atrium. All patients were in sinus rhythm.

Atrial size was assessed according to the recommendations for quantification of chambers [16] by different methods: anteroposterior diameter, biplanar area, and 2D biplanar volume. LA enlargement was considered when indexed volume was greater than 34 mL/m^2^.

The atrial function study included left atrial ejection fraction (LAEF) and speckle tracking. For the analysis of biplanar LAEF, we used the following formula: maximum volume − minimum volume/maximum volume × 100. GE Q-analysis application was used to analyse the strain. Speckle tracking echocardiography is a technique that uses acoustic back-scatter (speckles) generated by the reflected ultrasound beam. The displacement of this speckles is considered to follow myocardial movement and represents myocardial deformation. LA endocardial surface is manually traced by a point-and-click approach in four-chamber view, between the mitral annulus to the opposite mitral annulus side and excluding pulmonary veins and LA appendage. An epicardial surface tracing is then automatically generated by the system, thus creating a region of interest (ROI), and after manual adjustment of ROI, width the software divides into six segments. Lastly, the software analyses and generates longitudinal strain curves for each segment and a mean curve of all segments. The speckle analysis reflects the pathophysiology of the left atrium: the maximum systolic global longitudinal strain (PALs), corresponding to the atrial reservoir phase, and the atrial contraction strain (PACs), corresponding to the contractile pump phase [17] (Figure 1). The frame rate was maintained at a level > 50 frames/s, and onset of the QRS complex was used as a reference point (R-R gating).

### 2.3. Clinical Follow-Up

A clinical control was carried out in neurology and cardiology outpatients at 3, 6, and 12 months, together with a daily telematic follow-up. In case of detection of AF by the device, the registry was checked, and once the presence of AF was confirmed, anticoagulation treatment was started.

After this period, the patients continued with their telematic and standard clinical control in order to evaluate the recurrence of a new stroke and AF burden.

The study followed national and international guidelines (Declaration of Helsinki) and was approved by the local ethical committee CEIC-Parc de Salut Mar (code 2013/5055/I). All patients signed the informed consent form.

### 2.4. Statistical Analysis

The numerical variables were checked to follow a normal distribution using the Shapiro–Wilk and QQ-Plots tests, and their values are presented as mean ± standard deviation. Categorical variables are expressed by frequency and percentages. Differences between the two groups, with or without AF, were evaluated using the chi-squared test or Fisher’s test for categorical variables and Student’s *t*-test for independent samples for continuous variables.

Multivariate logistic regression analysis adjusted for age and sex was performed in several models. It included the variables indexed 2D volume, LAEF, PALs, and PACs in order to detect the association between them and the presence of AF in all patients. The area under the curve (AUC) of the receiver operating characteristic (ROC) analysis is presented for each model. To obtain the best cut-off points that discriminate the patients with the highest probability of presenting AF, we used bootstrapping analysis (with 1000 repetitions) for each of the following variables: indexed 2D volume, LAEF, PALs, and PACs, defined as the median of all points and the 2.5 and 97.5 percentiles as the confidence interval.

A value of *p* < 0.05 was considered statistically significant. Statistical software R version 3.4.2 (R: A Language and Environment for Statistical Computing. R Foundation for Statistical Computing, Vienna, Austria) was used.

## 3. Results

### 3.1. Demographic Characteristics and Atrial Fibrillation

During the study period, a total of 530 ischemic stroke patients between 50 and 89 years of age were seen at our hospital. After complete work-up during hospital stay, 80 patients were diagnosed with a cryptogenic stroke and met the eligibility criteria to participate in the study. During follow-up, five patients were ruled out for several reasons: complex aortic atherosclerotic plaques (*n* = 1), structural heart disease in echocardiogram (*n* = 2), or refusing insertable cardiac monitor (*n* = 2). Finally, the study group consisted of 75 patients with a mean age of 76 ± 9 years (56% male).

AF was detected in 37 (49%) patients during follow-up. The rate of AF detection after the stroke was 16 (21%) from implantation to 3 months, 11 (15%) at 6 months, and 10 (13%) at 12 months.

Initially, the AF was paroxysmal, but during the study follow up, AF recurrence was detected in 20 cases (temporal range from 2 min to 3 h), and AF became permanent in 11 cases.

No significant differences in demographic characteristics or risk factors were observed between patients with or without AF, with the exception of more advanced age in the group with AF (78 vs. 73 years, *p* = 0.047) (Table 1).

The mean follow-up was carried out for 58.5 months (interquartile range: 50.1–65.2). Patients diagnosed with AF and under anticoagulation treatment did not have any neurological ischemic event. On the other hand, of the group of patients who remained in sinus rhythm during the initial 12 months, only one patient was admitted for stroke together with the onset of AF at 13 months.

### 3.2. Atrial Characteristics and AF Detection

On univariate analysis, the biplanar area, the anteroposterior diameter, and the 2D atrial volume were higher in the group of patients with AF compared to the group without AF (*p* < 0.001). Likewise, the group of patients with AF showed impairment of LA contractile function with worse LAEF (46.8 ± 11.5% vs. 60.6 ± 5.2%, *p* < 0.001), reduced reservoir function (PALs: 19.6 ± 5.7% vs. 29.5 ± 7.2%, *p* < 0.001), and lower LA contraction (PACs: 8.9 ± 3.9% vs. 16.5 ± 6%, *p* < 0.001) (Table 1).

Multivariate logistic regression analysis adjusted for age and sex (Table 2) showed that LA volume or LAEF, as well as reservoir function (PALs) or atrial contraction (PACs), were associated with the detection of AF in patients with CrS. ROC curves analyses of these echocardiographic parameters showed a good sensitivity and specificity for detection of AF (Figure 2). After these results, bootstrapping analysis was performed to determine the optimal cut-off value of LA function parameters associated with AF (see Appendix A). These optimal thresholds related to AF were LAEF < 55%, PALs < 21.4%, and PACs < 12.9%.

### 3.3. Atrial Fibrillation Burden and Normal or Enlarged Left Atrium

As shown in Figure 3, 22 of the 75 (29%) patients had enlarged LA, and in 17 (77%) of them, AF was detected: paroxysmal AF in 10 and permanent in 7 cases. Furthermore, in 53 patients with normal LA volume, AF also presented in 20 (38%). Despite normal atrial size, myocardial LA strain analysis or LAEF detected the presence of atrial dysfunction in 29 patients, and in 18 (62%) of them, AF was diagnosed: paroxysmal—14 and permanent—4. In contrast, in 24 patients with normal atrial size and function, AF was only diagnosed in two (8%, *p* < 0.001) cases, both paroxysmal.

## 4. Discussion

The results from our study demonstrate the value of echocardiogram for detection of atrial disease in the setting of CrS using combined analysis of size and atrial function by 2D speckle tracking. Moreover, echocardiographic cut-off points were obtained in association with AF after CrS, with or without LA enlargement.

Atrial remodelling represents an alteration in atrial size and function secondary to cellular changes in the atrial tissue. Clinical observations suggest that several risk factors such as advanced age, arterial hypertension, and diabetes mellitus may lead to atrial remodelling with the development of interstitial fibrosis and increased atrial volume [18,19]. In clinical practice, these parameters can be assessed by using imaging techniques [20,21]. In particular, the degree of atrial fibrosis is inversely correlated with the value of the strain by echocardiography, a fact confirmed in histological studies [22].

Previous studies have demonstrated that LA enlargement by echocardiography is a strong predictor of AF [23]. However, the functional remodelling evaluated by reducing myocardial deformation has been shown to be more sensitive in detecting subclinical atrial dysfunction than atrial enlargement [24,25,26]. In this regard, retrospective studies performed in CrS have shown that speckle tracking echocardiography improved the diagnostic accuracy of AF in slightly enlarged atria [27,28]. These findings have been confirmed in the present study by multivariable regression: LA strain and LAEF showed an independent association with AF (Table 2). The area under the ROC curve (AUC) of LA volume taken alone was weaker at predicting AF than volume and atrial strain together, as shown in Figure 2. Kusunose et al demonstrated LA strain prediction value of AF in embolic stroke of undetermined source but without insertable loop recorder [29].

Furthermore, the optimal cut-off points obtained to predict AF were determined by bootstrapping analysis: LAEF < 55%, PALs < 21.4%, and PACs < 12.9%. Similar results have been reported in other contexts, such as predicting recurrence of AF after catheter ablation [30]. Silent AF was diagnosed in 38% (20 of 53) of patients with normal LA size, and myocardial analysis detected the presence of LA dysfunction in 29 patients; of them, in 18 (62%), AF was detected, 14—paroxysmal, and 4—permanent (Figure 3). In contrast, in 24 patients with normal atrial size and function, AF was only diagnosed in two (18%) patients. These results confirm previous findings described by Pagola et al. [5], who found AF in 86% of CrS patients with normal LA size and decreased LA strain in 33%.

These outcomes show the importance of routine atrial function assessment in clinical practice to better understand mechanisms of thrombogenesis after CrS due to the prognostic implications that it entails. Atrial hypocontractility results in abnormal stasis of blood within the atrium, supporting the emerging concept of atrial cardiomyopathy as a cause of stroke [31]. In the present study, patients who presented AF and were treated with oral anticoagulants, despite the recurrence of the arrhythmia, did not have another stroke.

Our results highlight a higher incidence of AF (49%) than that published in the literature [32]. This disagreement with previous reports may be due to different facts. Risk of AF increases exponentially with age, because AF is a disease of older adults and is usually associated with multiple comorbidities [33]. In our study, the mean age group was 76 years old, and 75% had arterial hypertension. In addition, the patients who presented AF were older than those without arrhythmia (78 vs. 73 years, *p* = 0.047). In most of the studies with CrS, the age of population included was usually under 70 years. On the other hand, continuous monitoring with an insertable cardiac monitor prior to hospital discharge allowed the diagnosis of 21% of AF during the first 3 months. In the CRYSTAL AF study [34], the inclusion of patients was carried out up to 90 days post-stroke, and the mean age of the patients was 61 years. In the SURPRISE study [35], the implantation of the insertable cardiac monitor was also long after the CrS (median of 69 days) and the study population was younger than ours. Moreover, a recent study has shown that the diagnosis of AF by using an insertable cardiac monitor compared to conventional strategy increased AF detection after CrS [36].

The limitations of our study require mention. Our study was carried out in a single centre with a small number of patients, although similar to that of previously published reports in the literature [37]. Future studies in a greater population should confirm our findings; in particular, the values of myocardial deformation, as a consequence of their variability. AF guidelines [38] define clinical AF as lasting at least 30 s; although the AF burden related outcomes are not clear, we consider the definition of ≥1 min’s duration, given that this cut-off value showed prognostic implications in a previous study after CrS [36].

## 5. Conclusions

Speckle tracking echocardiography is an extremely useful technique to improve the diagnosis of atrial disease in cryptogenic stroke. It could be of great assistance in stratifying the thromboembolic risk, especially in elderly patients with or without atrial enlargement.

## Figures and Tables

**Figure 1 jcm-10-03501-f001:**
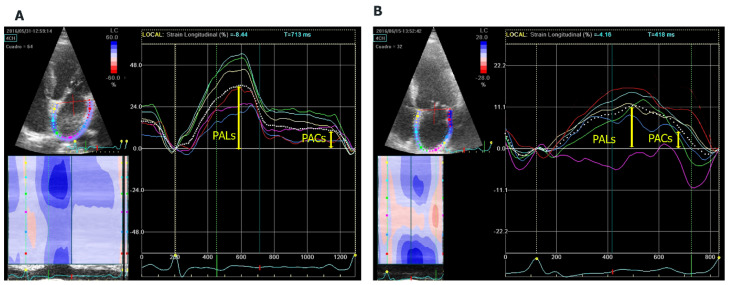
Analysis of left atrial myocardial deformation by 2D speckle tracking. (**A**) Four-chamber (4CH) apical longitudinal strain of a patient without AF with normal strain and (**B**) a patient with AF and reduced strain. PALs: peak atrial longitudinal strain. PACs: peak atrial contraction strain.

**Figure 2 jcm-10-03501-f002:**
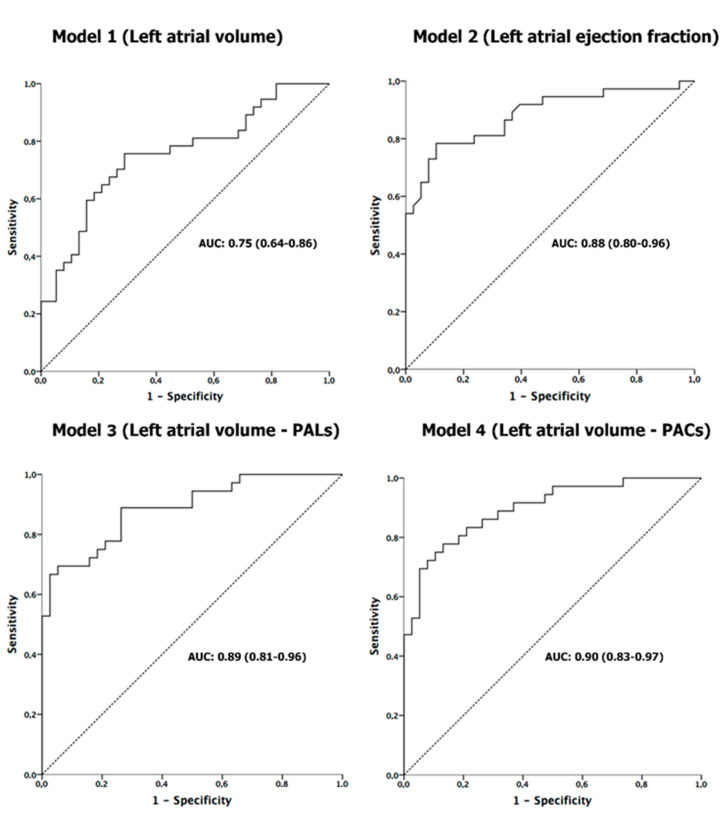
Results of receiver-operating characteristic curve analysis for predicting silent AF in all patients after CrS. AUC: area under the curve.

**Figure 3 jcm-10-03501-f003:**
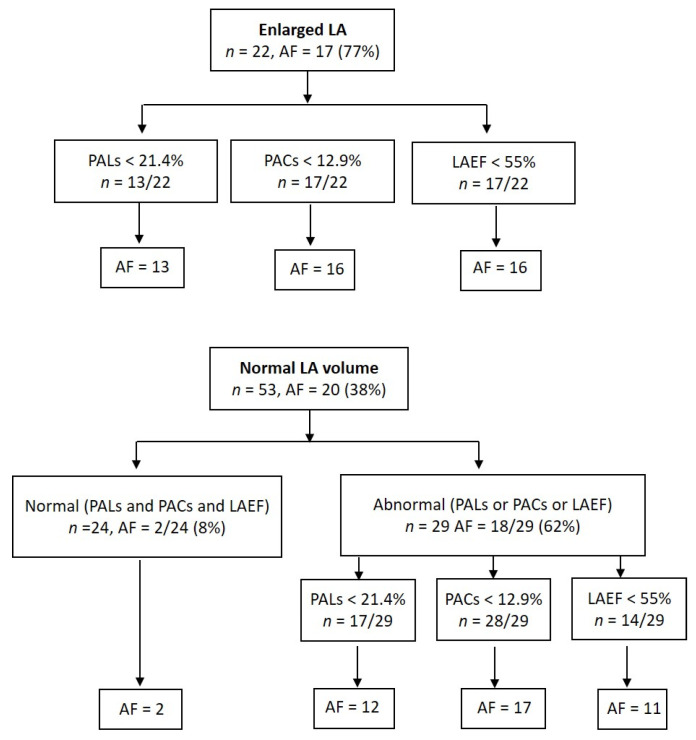
Tree diagram showing results of echocardiographic parameters according to whether left atrium was or was not enlarged.

**Table 1 jcm-10-03501-t001:** Clinical characteristics and echocardiographic parameters of patients with or without atrial fibrillation.

	Non AF (*n* = 38)	AF (*n* = 37)	*p*
Age	73.4 ± 9.7	77.6 ± 8.3	0.047
Gender (male)	23 (60.5%)	19 (51.4%)	0.570
Arterial hypertension	29 (76.3%)	28 (75.7%)	1.000
Diabetes	8 (21.1%)	9 (24.3%)	0.950
Dyslipidemia	21 (55.3%)	19 (51.4%)	0.914
Body mass index (kg/m^2^)	27 ± 4.14	28.1 ± 5.08	0.310
Glomerular filtration rate (mL/min)	76.4 ± 22.2	70.3 ± 26.2	0.286
Coronary artery disease	1 (2.63%)	3 (8.11%)	0.358
Lung disease	5 (13.2%)	4 (10.8%)	1.000
Previous stroke	6 (15.8%)	7 (18.9%)	0.958
CHA_2_DS_2_-VASc score	4.79 ± 1.43	5.22 ± 1.20	0.167
Echocardiogram			
LVEF (%)	63.2 ± 3.85	63.2 ± 2.83	0.996
LV mass (g)	172 ± 39.5	173 ± 56.3	0.924
Indexed LV mass (g/m^2^)	96.9 ± 19.6	98.1 ± 29.1	0.827
LA diameter (mm)	35 ± 4.03	38.5 ± 4.82	<0.001
LA area (cm^2^)	16.8 ± 2.55	19.8 ± 3.58	<0.001
2D LA volume (mL)	47.9 ± 12.5	60.8 ± 16.2	<0.001
Indexed 2D LA volume (mL/m^2^)	27 ± 6.49	34.7 ± 9.77	<0.001
Enlarged LA (>34 mL/m^2^)	5 (13%)	17 (46%)	
Normal LA volume	33 (87%)	20 (54%)	0.002
2D LAEF (%)	60.6 ± 5.2	46.8 ± 11.5	<0.001
PALs (%)	29.5 ± 7.24	19.6 ± 5.73	<0.001
PACs (%)	16.5 ± 6	8.99 ± 3.92	<0.001

LV: left ventricular; EF: ejection fraction; LA: left atrial.

**Table 2 jcm-10-03501-t002:** Multivariate logistic regression analysis models to detect atrial fibrillation after a cryptogenic stroke.

	OR	IC 95%	*p*
Model 1			
Age	1.04	0.98–1.11	0.216
Men	1.40	0.47–4.15	0.546
Left atrial volume	1.13	1.05–1.21	0.001
Model 2			
Age	1.04	0.96–1.12	0.342
Men	0.63	0.16–2.41	0.627
LAEF	0.80	0.72–0.89	<0.001
Model 3			
Age	1.01	0.94–1.09	0.783
Men	1.90	0.47–7.71	0.370
PALs	0.80	0.71–0.84	<0.001
Left atrial volume	1.12	1.02–1.23	0.013
Model 4			
Age	1.05	0.97–1.13	0.223
Men	1.69	0.44–6.48	0.446
PACs	0.72	0.59–0.87	0.001
Left atrial volume	1.10	1.00–1.20	0.057

LAEF: left atrial ejection fraction; PALs: peak atrial longitudinal strain; PACs: peak atrial contraction strain.

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
