# Peer review of "Left Atrium Assessment by Speckle Tracking Echocardiography in Cryptogenic Stroke: Seeking Silent Atrial Fibrillation"

_jcm, 2021, doi:10.3390/jcm10163501_

Round 1

Reviewer 1 Report

Ble et al present here a manuscript regarding left atrial assessment by 2D speckle tracking echocardiography performance in seeking silent atrial fibrillation after cryptogenic stroke. 

They found that left atrial volume, LAEF and LA strain in 4CV view were all independently associated with detection of AF diagnosed by loop Holter recording. Cut-off values of LAEF<55%, PALs<21.4% and PACs<12.9% showed increased risk of AF after CrS, reinforcing the potential for LA strain to improve silent atrial disease diagnosis after CrS, with our without LA enlargement.

Overall the subject is timely, of interest for the reader and well written. The authors should also be acknowledge for their interesting methodology approach, that will potentially target future research of larger magnitude.

Few comments:

1/ The methodological section of LA strain measurement needs to be slightly enhanced. The following publication can be helpful: Arch Cardiovasc Dis. 2021 Feb;114(2):96-104.

2/ The incremental power of LA function over LA size is demonstrated in other contexts (ex HCM, HFrEF). Referring to those can strengths the author's demonstration. See Circ Cardiovasc Imaging. 2021 Jan;14(1):e011608. 

3/ Figure 3 is difficult to understand. Those data could potentially be presented in a different way.

4/ Line 148 shows a typo "FA"

Author Response

DEAR REVIEWER:

Thank you very much for continuing to consider our manuscript entitled "Left atrium assessment by speckle tracking echocardiography in cryptogenic stroke: seeking silent atrial fibrillation".  We are grateful for reviewers’ and Editorial Office comments in this revision because they have helped us to improve the message of the paper. We respond to all comments point by point:

Ble et al present here a manuscript regarding left atrial assessment by 2D speckle tracking echocardiography performance in seeking silent atrial fibrillation after cryptogenic stroke. 

They found that left atrial volume, LAEF and LA strain in 4CV view were all independently associated with detection of AF diagnosed by loop Holter recording. Cut-off values of LAEF<55%, PALs<21.4% and PACs<12.9% showed increased risk of AF after CrS, reinforcing the potential for LA strain to improve silent atrial disease diagnosis after CrS, with our without LA enlargement.

Overall the subject is timely, of interest for the reader and well written. The authors should also be acknowledge for their interesting methodology approach, that will potentially target future research of larger magnitude.

Few comments:

1/ The methodological section of LA strain measurement needs to be slightly enhanced. The following publication can be helpful: Arch Cardiovasc Dis. 2021 Feb;114(2):96-104.

Reply: The description of LA strain acquisition has been modified and this study has been mentioned in the text.

2/ The incremental power of LA function over LA size is demonstrated in other contexts (ex HCM, HFrEF). Referring to those can strengths the author's demonstration. See Circ Cardiovasc Imaging. 2021 Jan;14(1):e01160

Reply: The second study suggested demonstrated relationship between left atrial volume and myocardial velocity at atrial contraction by pulsed Doppler tissue imaging (DTI) that carries prognostic significance for survival in patients with heart failure with reduced ejection fraction. Although the article is interesting, the study population is different from that of our study. In addition, echocardiographic quantification of regional myocardial function of left atrial was assessed by DTI (angle-dependent) and in our study by speckle-tracking echocardiographic techniques (angle-independent).

3/ Figure 3 is difficult to understand. Those data could potentially be presented in a different way.

Reply: This figure has been modified.

4/ Line 148 shows a typo "FA"

Reply: We corrected the grammar error.

Reviewer 2 Report

The authors analyzed atrial size and function by speckle tracking echocardiography in CrS patients to detect atrial disease. They measured left atrial ejection fraction (LAEF) and atrial strain. Subcutaneous implantable cardiac monitors were also implanted in these patients. Seventy-five consecutive patients were included, aged 76±9 years and AF was diagnosed in 49% of cases. The AF group had higher atrial volume and worse atrial function: peak atrial longitudinal strain (PALs) 19.6±5.7% vs.29.5±7.2%, peak atrial contraction strain (PACs) 8.9±3.9% vs.16.5±6%; LAEF (46.8±11.5% vs.60.6±5.2%) p<0.001. AF was diagnosed in 20 of 53 patients with non-enlarged atrium and in 18 of them atrial dysfunction was present. The multivariate logistic regression analysis demonstrated an independent association between detection of AF and atrial volume, LAEF and strain. The authors demonstrated cut-off values LAEF<55%, PALs<21.4% and PACs<12.9%. The authors concluded that speckle tracking echocardiography in CrS patients improves silent atrial disease diagnosis, with or without atrial enlargement. The manuscript is overall well written and the findings are of interest. I have several comments for the authors to address:

  1. The findings are not novel. These findings have previously been published by other authors in the context of atrial fibrillation. (Kusunose K, Takahashi H, Nishio S, Hirata Y, Zheng R, Ise T, Yamaguchi K, Yagi S, Fukuda D, Yamada H, Soeki T, Wakatsuki T, Shimada K, Kanematsu Y, Takagi Y, Sata M. Predictive value of left atrial function for latent paroxysmal atrial fibrillation as the cause of embolic stroke of undetermined source. J Cardiol. 2021 Jun 9:S0914-5087(21)00129-5. doi: 10.1016/j.jjcc.2021.05.005. Epub ahead of print. PMID: 34119401.) (Wen S, Indrabhinduwat M, Brady PA, Pislaru C, Miller FA, Ammash NM, Nkomo VT, Padang R, Pislaru SV, Lin G. Post Procedural Peak Left Atrial Contraction Strain Predicts Recurrence of Arrhythmia after Catheter Ablation of Atrial Fibrillation. Cardiovasc Ultrasound. 2021 Jun 11;19(1):22. doi: 10.1186/s12947-021-00250-5. PMID: 34116696; PMCID: PMC8194218.)
  2. It appears from the data that the left atrial diameter measured higher in those with atrial fibrillation. What added advantage does measurement of left atrial ejection fraction, peak atrial contraction strain and peak atrial longitudinal strain provide over simple measurement of left atrial diameter? What was the AUC for left atrial diameter?

Author Response

DEAR REVIEWER:

Thank you very much for continuing to consider our manuscript entitled "Left atrium assessment by speckle tracking echocardiography in cryptogenic stroke: seeking silent atrial fibrillation".  We are grateful for reviewers’ and Editorial Office comments in this revision because they have helped us to improve the message of the paper. We respond to all comments point by point.

The authors analyzed atrial size and function by speckle tracking echocardiography in CrS patients to detect atrial disease. They measured left atrial ejection fraction (LAEF) and atrial strain. Subcutaneous implantable cardiac monitors were also implanted in these patients. Seventy-five consecutive patients were included, aged 76±9 years and AF was diagnosed in 49% of cases. The AF group had higher atrial volume and worse atrial function: peak atrial longitudinal strain (PALs) 19.6±5.7% vs.29.5±7.2%, peak atrial contraction strain (PACs) 8.9±3.9% vs.16.5±6%; LAEF (46.8±11.5% vs.60.6±5.2%) p<0.001. AF was diagnosed in 20 of 53 patients with non-enlarged atrium and in 18 of them atrial dysfunction was present. The multivariate logistic regression analysis demonstrated an independent association between detection of AF and atrial volume, LAEF and strain. The authors demonstrated cut-off values LAEF<55%, PALs<21.4% and PACs<12.9%. The authors concluded that speckle tracking echocardiography in CrS patients improves silent atrial disease diagnosis, with or without atrial enlargement. The manuscript is overall well written and the findings are of interest. I have several comments for the authors to address:

  1. The findings are not novel. These findings have previously been published by other authors in the context of atrial fibrillation. (Kusunose K, Takahashi H, Nishio S, Hirata Y, Zheng R, Ise T, Yamaguchi K, Yagi S, Fukuda D, Yamada H, Soeki T, Wakatsuki T, Shimada K, Kanematsu Y, Takagi Y, Sata M. Predictive value of left atrial function for latent paroxysmal atrial fibrillation as the cause of embolic stroke of undetermined source. J Cardiol. 2021 Jun 9:S0914-5087(21)00129-5. doi: 10.1016/j.jjcc.2021.05.005. Epub ahead of print. PMID: 34119401.) (Wen S, Indrabhinduwat M, Brady PA, Pislaru C, Miller FA, Ammash NM, Nkomo VT, Padang R, Pislaru SV, Lin G. Post Procedural Peak Left Atrial Contraction Strain Predicts Recurrence of Arrhythmia after Catheter Ablation of Atrial Fibrillation. Cardiovasc Ultrasound. 2021 Jun 11;19(1):22. doi: 10.1186/s12947-021-00250-5. PMID: 34116696; PMCID: PMC8194218.)

Reply:

The first article, Kusunose K et al (J Cardiol. 2021 Jun 9: S0914-5087 (21) 00129-5) confirmed that LA strain is higher than LA volume (AUC 0.76 + 0.05) vs AUC 0.68 + 0, 04, p = 0.05), respectively in predicting AF in embolic stroke of undetermined source. In this regard the present study would not be new. However, our study presents the following differences:

  • Longer follow-up (mean 58.5 months) of patients who have been monitored during admission and with an insertable cardiac monitor prior to discharge. In the study by Kusunose K et al, the follow-up was shorter with a median of 18 days in admitted patients.
  • In our study, the left atrium was divided into normal-sized atria and enlarged atria, according to the atrial volume recommended by the guidelines. Detection of AF on patients with normal LA volume has been possible by LA strain or LA ejection fraction. This fact is clinically relevant, because these patients under coagulation treatment did not have any recurrence of ictus. Moreover, the medical resources are limited, but the identification of high-risk populations leads to more effective use of medical management in the clinical setting.
  • Finally, a cut-off point has been described using the bootstrapping technique that should be verified in future studies in this population.

The second article, Wen S et al (Cardiovasc Ultrasound. 2021) studied a population different from ours. The population studied was AF undergoing catheter ablation. The authors demonstrate that the <12% cut-off point of peak left atrial contraction (PACs) is a predictor of AF recurrence. This is in agreement with our results, where the cut-off point of the LA PACs in cryptogenic stroke patients was also a predictor of recurrence.

Both studies support the results of the present study and have been mentioned in the text.

  1. It appears from the data that the left atrial diameter measured higher in those with atrial fibrillation. What added advantage does measurement of left atrial ejection fraction, peak atrial contraction strain and peak atrial longitudinal strain provide over simple measurement of left atrial diameter? What was the AUC for left atrial diameter?

Reply:

It is true that the LA diameter is significantly higher in patients with atrial fibrillation. Although the logistic regression analysis was significant (OR 1.24 95% CI 1.040-1.482) and the AUC curve was 0.71 (0.59-0.89). However, assessment of LA size using only the PA diameter assumes that when the left atrium is enlarged, all of its dimensions change similarly, which is often not the case during LA remodeling. In contrast, LA volume takes into account LA chamber size alterations in all directions.

On the other hand, the ease with which LA volumes can be obtained in clinical practice, together with the existing solid literature on normal values and their prognostic value, lead to the guidelines recommending the measurement of LA volume, instead of the diameter. (Lang; 2019 guides). Moreover, guidelines provide the upper normal limit for 2D echocardiographic LA volume is 34 mL/m2 for both genders.

Round 2

Reviewer 2 Report

The authors have addressed my comments satisfactorily.